# A Case Study Assessing the Liquefaction Hazards of Silt Sediments Based on the Horizontal-to-Vertical Spectral Ratio Method

Qingsheng Meng [1,2,3,*], Yang Li [1,*], Wenjing Wang [1], Yuhong Chen [1] and Shilin Wang [1]

1 College of Environmental Science and Engineering, Ocean University of China, Qingdao 266100, China
2 Key Laboratory of Marine Environment Science and Ecology, Ministry of Education, Qingdao 266100, China
3 Key Laboratory of Marine Environmental Geological Engineering, Qingdao 266100, China
* Correspondence: qingsheng@ouc.edu.cn (Q.M.); liyang4009@stu.ouc.edu.cn (Y.L.);
  Tel.: +86-138-5327-4653 (Q.M.); +86-176-6090-8189 (Y.L.)

**Abstract:** Silt liquefaction can occur due to the rapid cyclic loading of sediments. This can result in the loss of the bearing capacity of the underlying sediments and damage to the foundations and infrastructure. Therefore, assessing liquefaction hazards is an important aspect of disaster prevention and risk assessment in geologically unstable areas. The purpose of this study is to assess the liquefaction hazards of silt sediments by using the horizontal-to-vertical spectral ratio method. Single-station noise recording was carried out in the northern plain of the Yellow River Delta, and a new method was adopted to identify the fundamental frequency. The dynamic parameters of the silt, such as the fundamental frequency, amplification, and vulnerability index, were used as indicators to assess the liquefaction potential. The results show that the silty soils in different areas have different stable ranges of values of the fundamental frequency. Moreover, the distribution of the observations is in good agreement with the geological conditions in the area, which indicates the potential applicability and reliability of the new method for identifying fundamental frequency. The vulnerability index is inversely related to the fundamental frequency, with the southwestern part of the study area having a lower fundamental frequency and a higher vulnerability index, meaning a greater liquefaction risk compared to other areas. The horizontal-to-vertical spectral ratio method has great advantages in characterizing subsurface dynamic parameters and can be applied to liquefaction hazard assessments of silt sediments in large areas, which is critically important in terms of providing information and guidance for urban construction and planning.

**Keywords:** HVSR; liquefaction potential; fundamental frequency; amplification; vulnerability index

## 1. Introduction

Continuous vibrations, such as from earthquakes or strong storms, can cause loosely deposited sediments on or near the surface to undergo liquefaction. Liquefaction is a common and harmful form of unstable failure of soil and is a scientific issue of widespread interest among the domestic and international geotechnical and earthquake engineering communities. When liquefaction occurs, the stability of the soil skeletal is disrupted, and the soil particles take on a free state. This causes the soil to exhibit fluid-like properties, which loses its bearing capacity and results in damage to overlying structures. For example, the 1964 Niigata earthquake in Japan [1], the 1994 Northridge earthquake in California [2], the 1999 Chi-Chi earthquake in China [3], and the 2011 Christchurch earthquake in New Zealand [4] all caused severe damage to buildings, embankments, roads, and underground facilities due to soil liquefaction.

There are many factors that affect soil liquefaction, and real liquefaction events are rare and difficult to reproduce. As a result, liquefaction hazard assessments are critical for preventing these hazards. At present, assessing liquefaction potential primarily adopts

laboratory tests and field tests [5]. The dynamic triaxial test [6] uses stress and strain of soil as the criteria to assess liquefaction. The standard penetration test (SPT) [7], the piezocone penetration test (PCPT) [8,9], and geophysical tests [10,11] are used to directly characterize the liquefaction potential of soils based on data from field investigations. Due to their safety and reliability, the above methods have been successfully applied to the study of sand liquefaction [12–15]. However, SPT and CPT are not suitable for large areas because of their complex implementation and high cost. Furthermore, laboratory tests cannot guarantee the structural characteristics of in situ soils; thus, they cannot truly reflect the dynamic response of sediments. Moreover, most studies have only focused on sandy soils, and no consensus has been reached on liquefaction assessments of fine-grained sediments containing silt, silty sand, and silty clay. Unlike sandy soils, silty soils are dominated by fine particles such as silt, fine sand, and small amounts of interspersed clay. With weaker permeability and poorer water stability, these sediments are prone to uneven settlement. Therefore, it is of great practical importance to assess the liquefaction potential of silty soil during local engineering and construction projects, and to ensure the safety of life and property.

Ambient noise is a kind of vibration caused by natural or cultural origins such as waves, tides, the effects of wind, industrial machinery, and so on. The horizontal-to-vertical spectral ratio (HVSR) method is based on ambient noise recordings and is a key technique proposed by Nakamura [16] to determine the dynamic response of soft sediment [17,18]. This method only requires a three-component geophone to quickly investigate the dynamic characteristics of undisturbed soil over a large area by measuring properties such as fundamental frequency ($f_0$) and amplification ($A$). Due to these advantages, HVSR has been rapidly developing in the fields of site effect evaluation [19,20], seismic microzonation [21,22], and evaluation of sedimentary cover thickness [23,24]. In the past, the frequency that corresponded to the peak amplitude of the HVSR curve was considered to be the fundamental frequency. However, the spectrum of the actual recording has multiple peaks that Nakamura [25,26] described as being caused by multiple reflections of the vertically incident SH waves. Moreover, the energy of the fundamental Rayleigh waves in the noise wavefield generates additional peak amplitudes on the HVSR curve [27], which interferes with the identification of the fundamental frequency. Therefore, it is necessary to find a more suitable method for identifying the fundamental frequency.

The purpose of this paper is to assess silt liquefaction hazards using the HVSR method. We established a method to identify the fundamental frequency based on the variable differences of the amplitude spectrum between the horizontal and vertical ambient noise components and conducted single-station noise recording tests at multiple sites in the Yellow River Delta plain. Then, the HVSR method was used to process the data to obtain the fundamental frequency and amplification for each station, and the vulnerability index ($K_g$) was calculated to estimate the potential liquefaction sites. The liquefaction hazards in the Yellow River Delta are evaluated based on geological information and the distribution of the fundamental frequency and amplification and the vulnerability index.

## 2. Methodology

The HVSR method is a passive source seismic technique that requires only a single three-component geophone to record ambient noise. Fourier spectrum analysis is then performed for each component of the ambient noise, converting the time-domain data to frequency-domain data. The method analyzes the spectral ratio of the Fourier amplitudes of the horizontal and vertical noise components, from which the HVSR curves and fundamental frequencies of the stations are then estimated [28].

$$\text{HVSR} = \frac{\sqrt{(\text{H}_{EW}(f)^2 + \text{H}_{EW}(f)^2)/2}}{\text{H}_V(f)} \tag{1}$$

where $\text{H}_{EW}(f)$ is the Fourier amplitude spectrum of the east–west component of the noise signal, $\text{H}_{NS}(f)$ is the Fourier amplitude spectrum of the north–south component and $\text{H}_V(f)$ is the Fourier amplitude spectrum of the vertical component. Fourier amplitude

spectra can understand the frequency components of ambient noise and be used for hazard analysis. In addition, several ground motion parameters, such as fundamental frequency and amplification can be obtained from the Fourier amplitude spectrum.

The HVSR method can estimate the characteristics of ground motion by recording ambient noise at the surface. The HVSR curve is plotted based on the ratio of the Fourier amplitude spectra of the horizontal and vertical components of the ambient noise, which is usually related to the wave impedance. By determining the fundamental frequency of the site and the corresponding spectral amplitude, the amplification characteristics of the actual soil can be estimated. Based on this premise, Nakamura [29] proposed a vulnerability index, to estimate the risk of soil damage according to the HVSR results.

$$K_g = \frac{A^2}{f_0} \qquad (2)$$

where $K_g$ means the vulnerability indices of ground, and $g$ means ground. $A$ is the amplification and $f_0$ is the fundamental frequency. The vulnerability index ($K_g$) value is derived from the ground motion parameters and is related to the strain on the ground and the basement. Therefore, $K_g$ can be used to characterize the easiness of deformation at the measurement point, which can be used to detect ground weak points.

## 3. Application in the Yellow River Delta

### 3.1. Geologic Setting

As shown in Figure 1, the study area is located in the northern part of the Yellow River Delta plain, which has an overall gentle terrain. The Yellow River Delta is a typical fast-depositing delta, with a characteristic dual sedimentary structure consisting of river alluvium covering the marine layer [30,31]. The overlying sediments were formed by rapid scouring and siltation driven by the hydrodynamic forces and coastal sedimentation processes. These sediments are mainly composed of silt and silty clay, and become thinner from southwest to northeast. Due to the high sand content, high water content, high liquidity index, and poor grading, the silt has low strength and a low bearing capacity. Additionally, during the continuous deposition of the deltaic strata, soft soils have been generally developed and are unevenly distributed. These conditions may lead to local geohazards in the soil when subjected to dynamic loadings, such as by uneven soil settlement or liquefaction.

### 3.2. Instruments and Field Investigation

In the field test, the equipment used for ambient noise recording during the field investigation included a WZG-6A engineering seismograph for data acquisition, a PS-2B three-component velocity geophone with a dominant frequency of 2 Hz, and a compass, as shown in Figure 2.

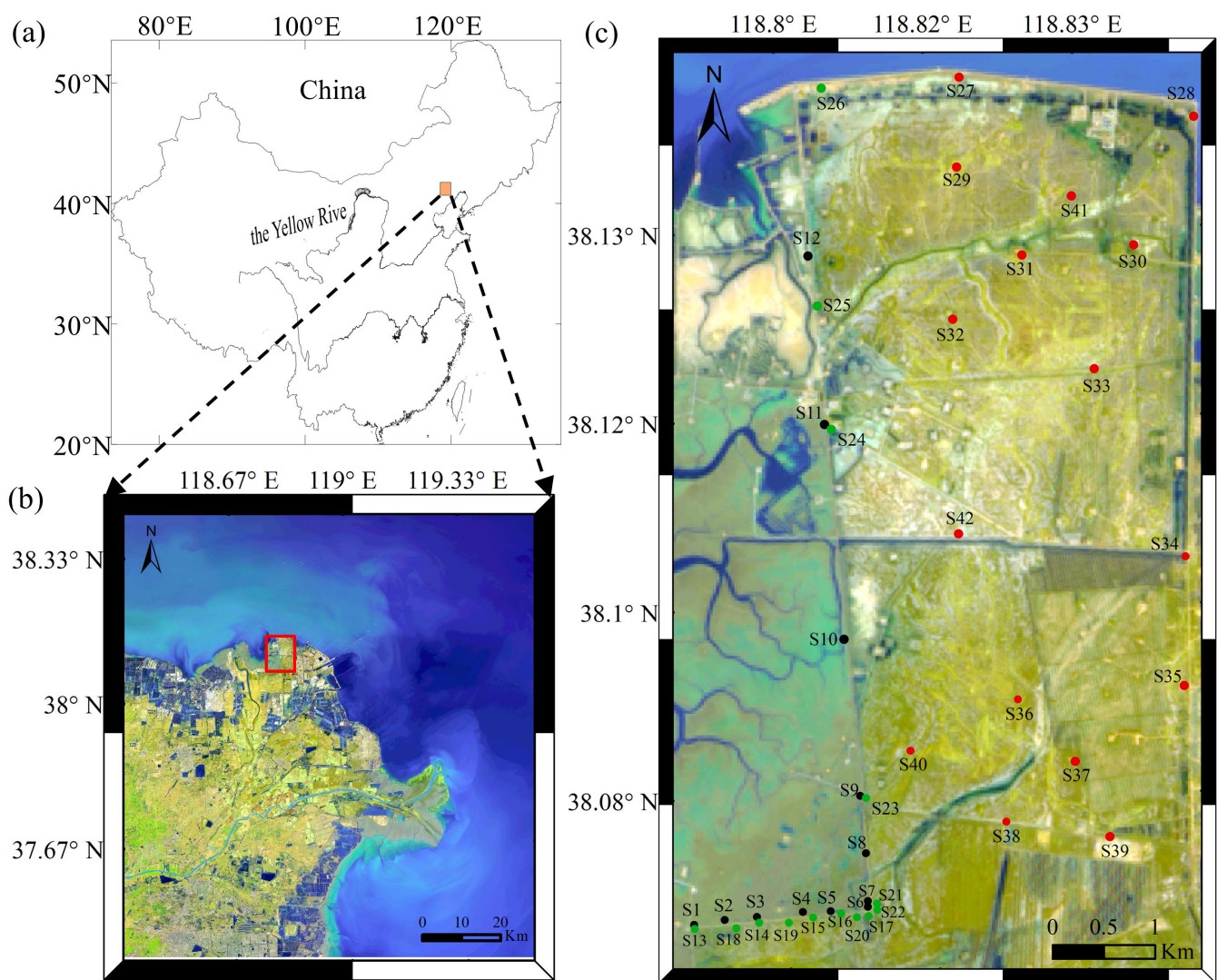

**Figure 1.** (**a**) Site of the Yellow River Delta (orange rectangle). (**b**) Map of the study area (red rectangle). (**c**) Distribution of measurement stations within the study area.

The study area is divided into three zones based on the location of the stations in Figure 1, of which S1–S12 are located in the beach area in front of the embankment (BFE), S13–S26 are located in the area behind the embankment (ABE), and S27–S42 are located in the wetland in the subaerial delta (WISD). The first step of the test was to determine the location of the observation station by GPS and the north direction by compass. The second step was to set up the three-component geophone. The geophones were installed directly on the cleared ground or in a hole (approximately 40 cm deep), oriented so that the *X*-axis pointed east and the *Y*-axis pointed north, and then leveled. The main purpose of installing the geophone in a hole was to enhance the coupling between the geophone and the soil and to minimize the effects of bad weather. Finally, the status of the device connection was checked, and the recording at each station had a duration of approximately 3 min, with a sampling frequency of 100 Hz. During acquisition, it is essential to avoid interference from transient strong signals caused by vehicles and footsteps, and so on, and to ensure that the geophone remains sufficiently coupled to the soil [32,33].

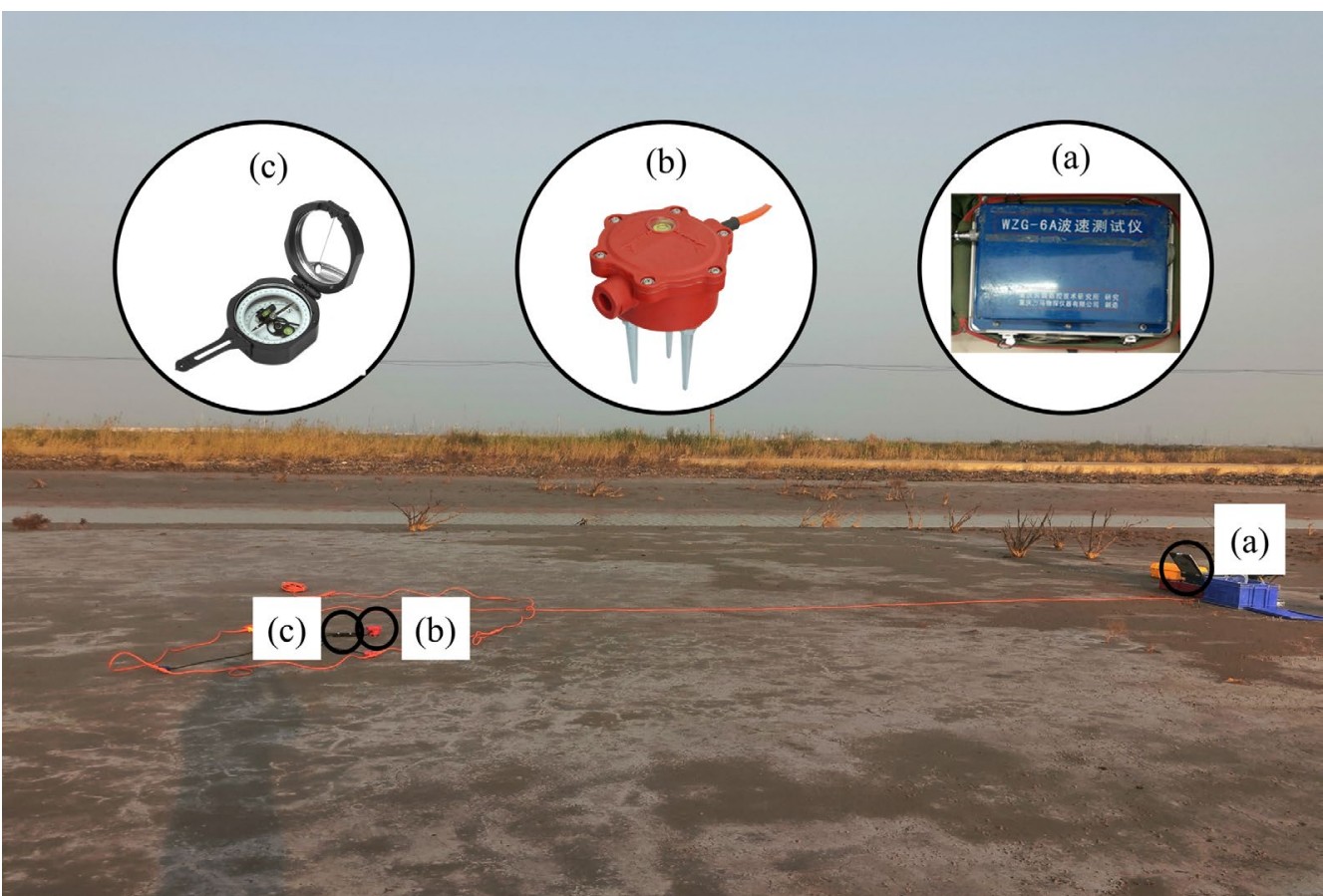

**Figure 2.** Pictures of the field test. (**a**) Data acquisition instrument. (**b**) Three-component geophone. (**c**) Compass.

### 3.3. Data Processing

In this study, we used the "OpenHVSR-Processing toolkit" computer program to process the recorded signals. This program is a new, interactive visualization tool for the HVSR processing of large noise datasets, which can highlight the spatial correlation of noise [34]. The calculation steps of the HVSR were shown in Figure 3. In the first data processing step, the continuously recorded three-component noise data are filtered and split into a stationary window of 10 s, with STA/LTA of 2.0 and overlap of 50%. This is to eliminate the interference of noise deviating from the normal distribution caused by transient impulse noise (e.g., vehicles passing near the geophone). In the second step, the Fourier amplitude spectrum for each time window is calculated and smoothed using the Konno and Ohmachi [35] method with a bandwidth coefficient (b) of 40, and the window's cosine tapering value of 5%. The subsequent step is to average the Fourier spectra of the two horizontal components and calculate the H/V of each window. Finally, the average HVSR of all windows is calculated.

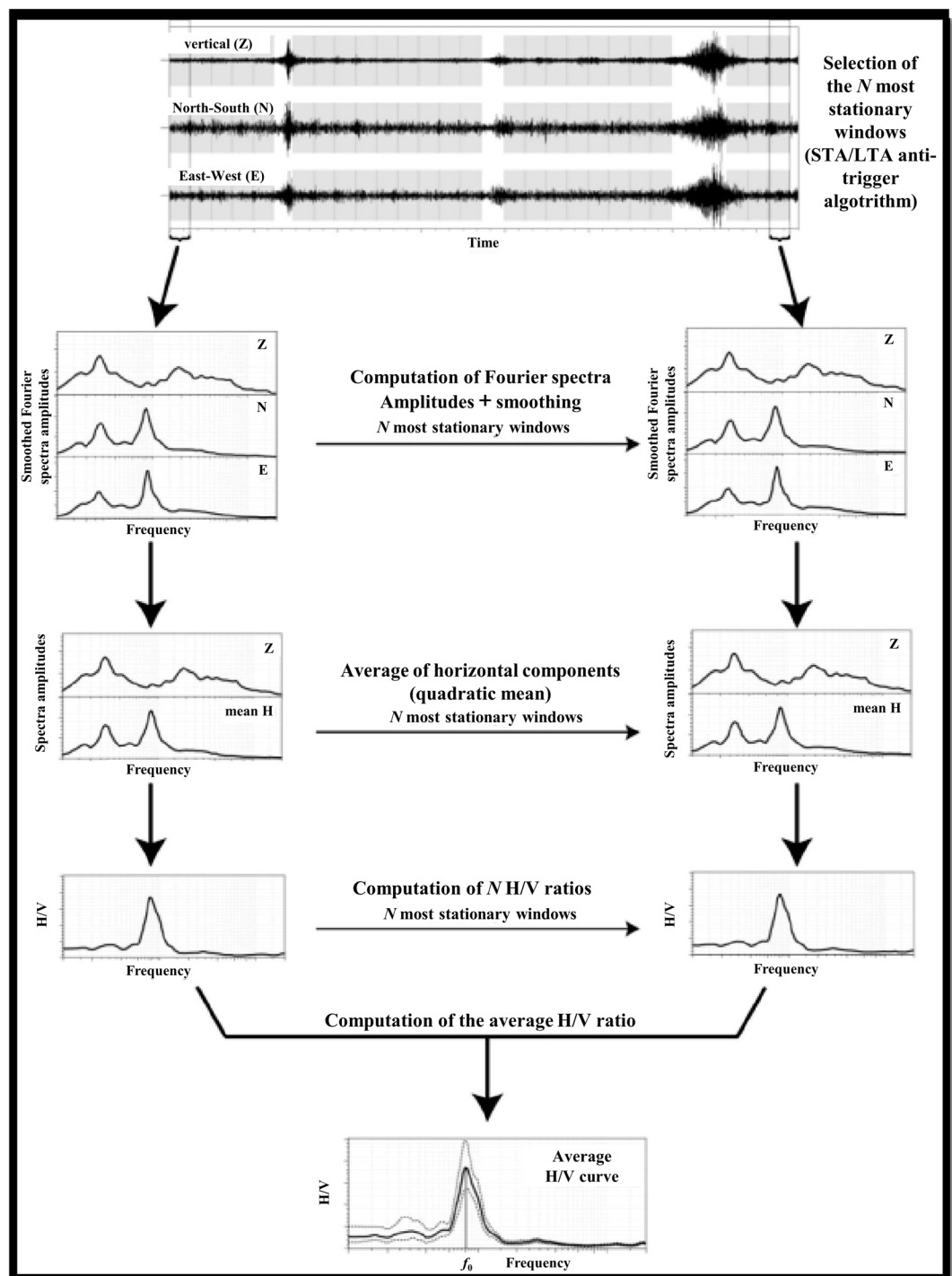

**Figure 3.** Flow chart shows the calculation steps of the HVSR. (http://www.geopsy.org/documentation/geopsy/hv.html) "URL (accessed on 10 November 2022)".

### 3.4. Identify Fundamental Frequency

It was shown that the fundamental frequency identified by HVSR coincides with the resonant frequency of the shear waves in the surface sediments, irrespective of the nature of the noise wavefield [36]; thus, it is a crucial step to correctly identify the fundamental frequency. Nakamura [25–27] argued that when seismic waves propagate through firm and homogeneous strata, the vibrations are uniform in each frequency range and each direction and do not amplify the amplitude. However, when seismic waves are vertically incident

to the firm strata and propagate into the overlying loose sediment strata, the horizontal motion of the seismic waves is amplified but the vertical motion is barely amplified.

The amplification of seismic waves is related to factors such as density variations between strata, P-wave velocity, S-wave velocity, and layer thickness. The HVSR method is based on monitoring the amplification of shear waves on the surface. Therefore, I propose a new method to identify the fundamental frequency based on the difference in amplification of vertically incident S-waves. This method determines the fundamental frequency by comparing the frequency bands where the horizontal component signal is enhanced but the vertical component is not. When noise propagates in the ground, vibrations in the noise wavefield with the same frequency as the resonant band of the ground will cause a resonant response in the soil. The corresponding ground motion with the strongest energy of seismic waves can be detected. The amplitude of the horizontal component of seismic waves is amplified when the seismic waves are incident vertically from the hard strata into the soft strata. Therefore, the fundamental frequency is accurately identified by monitoring the horizontal and vertical components of ground motion at the surface and considering the energy distribution of the Fourier spectrum.

Figure 4 shows the processing results for the S1 stations. Figure 4a shows the record of the V, E–W, and N–S noise components after filtering and windowing. The white windows are those that were filtered or cleaned because they contained transient signals. It shows the tiled views of the spectral ratios for H/V, E/V, and N/V, respectively, in Figure 4b, and the tiled views of the Fourier spectra for each window of the three components in Figure 4c. As shown in Figure 4d, the mean HVSR and 95% confidence intervals for all windows are shown in the left panel, the comparison of the mean curves labeled H/V, E/V, and N/V are presented in the middle, and the smoothed amplitude spectral curves for the E, N, and V components are shown in the right panel. Finally, Figure 4e shows the spectral ratios for H/V, E/V, and N/V respectively. It can be seen from Figure 4, the amplitude spectrum of each component shows that the E–W and N–S components begin to increase from 1.0 Hz, while the vertical component barely increases in the 1.0–1.85 Hz range. Furthermore, according to the Fourier spectrum of the N–S component, the vertical component starts to increase at approximately 1.8 Hz, and the tiled views of the spectral ratio are at their maximum at that frequency. Therefore, 1.8 Hz was identified as the fundamental frequency of the soil at this station.

The results of the identification of fundamental frequency using the method of this paper and another method, as shown in Figure 5. The previous method finds the fundamental frequency by determining the peak amplitude, as the green line in Figure 5a. Nevertheless, there are obviously multiple peaks in the HVSR curve. Combined with the observation data and local geological conditions, the site is located in the beach area in front of the embankment, which could not have such a high value of fundamental frequency. The previous method may have some errors when applied in the actual complex surroundings. Thus, we use the method proposed in this paper for identification. As can be seen in Figure 5c, the vertical component of the noise is barely amplified in the range of 0.8–1.6 Hz, while the horizontal component of the noise is amplified. The HVSR curve has a peak at 1.4 Hz, as shown in the red line in Figure 5a. Moreover, the H/V, E/V, and N/V ratios have peaks here in Figure 5b. Therefore, we conclude that the value of the fundamental frequency at this station is 1.4 Hz.

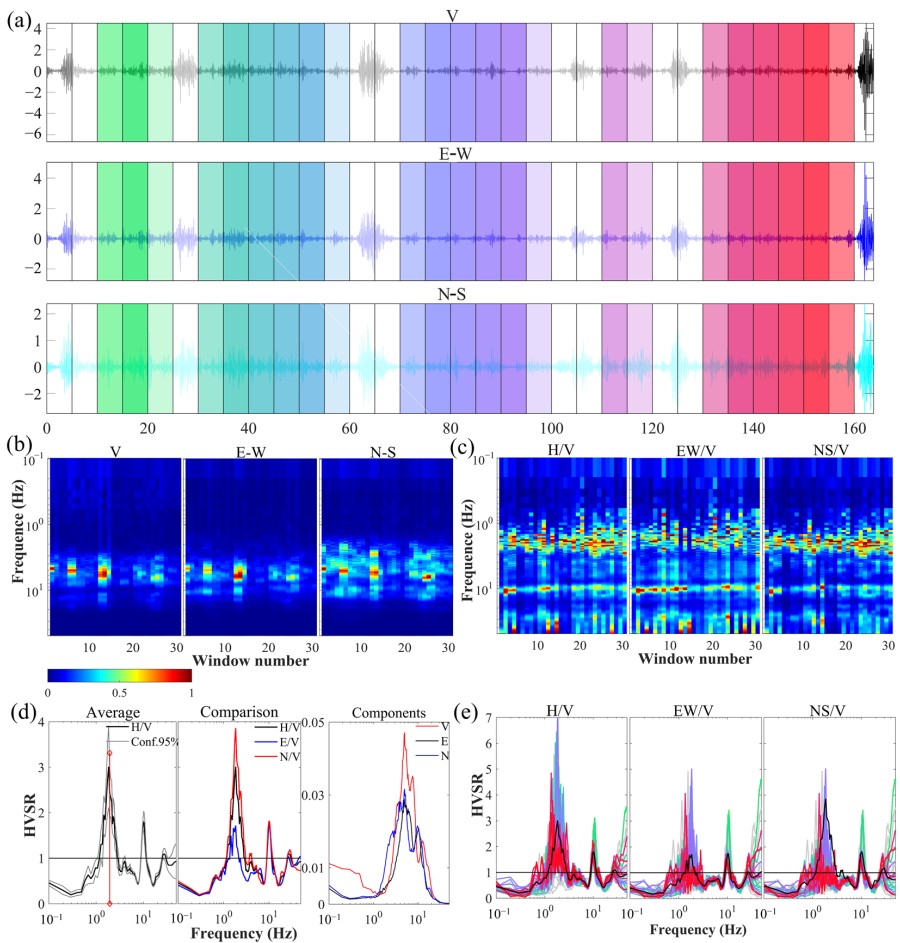

**Figure 4.** Processing result for S1. (**a**) Windowing. (**b**) Tiled view of Fourier spectra. (**c**) Tiled view of spectral ratios. (**d**) Average HVSR curve (left), H/V, E/V, and N/V ratios (center), the smoothed E, N, and V spectra (right). (**e**) HVSR curves for all windows, H/V, E/V, and N/V are shown from left to right.

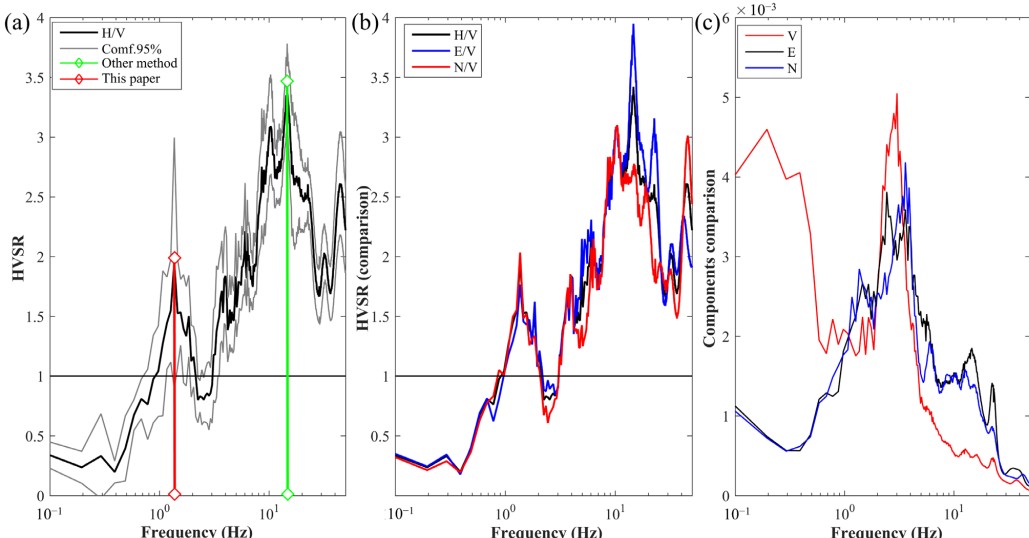

**Figure 5.** The identification of the fundamental frequency of S9 using the method of this paper and other method. (**a**) Average HVSR curve. (**b**) The H/V, E/V, and N/V ratios. (**c**) The smoothed E, N, and V spectra.

## 4. Discussion

To evaluate the liquefaction hazard in the Yellow River Delta, the ambient noise recordings from 42 stations were processed to calculate the HVSR, of which 31 results met the criteria for a reliable HVSR curve as defined by SESAME [37]. The results are shown in Table 1. The poor results from the other 11 stations may have been caused by poor coupling of the instrument to the soil or by severe weather conditions.

**Table 1.** The results of ambient noise recording from the Yellow River Delta.

| ID | Longitude | Latitude | Altitude | $f_0$ (Hz) | $A$ | $K_g$ | Number of Stationary Window |
|---|---|---|---|---|---|---|---|
| S1 | 4,226,376 | 920,357 | 0.2 | 1.8 | 3.0 | 5.0 | 23 |
| S2 | 4,226,434 | 920,654 | −0.1 | 1.5 | 1.8 | 2.2 | 19 |
| S3 | 4,226,474 | 920,970 | 0.0 | 3.7 | 2.1 | 1.2 | 20 |
| S4 | 4,216,312 | 394,829 | 0.0 | 1.4 | 2.5 | 4.5 | 21 |
| S5 | 4,226,555 | 921,692 | 0.0 | 1.5 | 2.1 | 2.9 | 22 |
| S6 | 4,226,608 | 922,055 | 0.3 | 1.4 | 2.1 | 3.2 | 25 |
| S7 | 4,216,401 | 395,467 | 0.0 | 1.3 | 2.0 | 3.1 | 32 |
| S8 | 4,227,136 | 922,019 | 0.6 | 0.8 | 3.5 | 15.3 | 28 |
| S9 | 4,227,697 | 921,944 | 0.3 | 1.4 | 1.9 | 2.6 | 23 |
| S10 | 4,229,229 | 921,736 | 0.0 | N/A | N/A | N/A | 19 |
| S11 | 4,231,335 | 921,477 | 0.3 | N/A | N/A | N/A | 8 |
| S12 | 4,232,986 | 921,262 | 0.4 | 1.5 | 1.8 | 2.2 | 22 |
| S13 | 4,226,335 | 920,364 | 1.2 | N/A | N/A | N/A | 20 |
| S14 | 4,226,417 | 920,992 | 1.1 | 3.7 | 2.0 | 1.1 | 28 |
| S15 | 4,226,485 | 921,521 | 1.3 | 3.8 | 2.8 | 2.1 | 26 |
| S16 | 4,226,524 | 921,809 | 1.1 | 3.7 | 3.2 | 2.8 | 26 |
| S17 | 4,226,518 | 922,066 | 1.0 | 3.7 | 2.5 | 1.7 | 24 |
| S18 | 4,216,175 | 394,172 | 1.3 | 3.8 | 2.5 | 1.6 | 22 |
| S19 | 4,216,212 | 394,689 | 1.2 | 3.8 | 2.3 | 1.4 | 28 |
| S20 | 4,216,244 | 395,354 | 1.1 | 4.0 | 2.5 | 1.6 | 22 |
| S21 | 4,216,373 | 395,557 | 1.2 | 3.7 | 3.0 | 2.4 | 23 |
| S22 | 4,226,579 | 922,147 | 0.8 | 3.7 | 3.0 | 2.4 | 29 |
| S23 | 4,227,677 | 922,001 | 0.9 | 1.4 | 3.0 | 6.4 | 23 |
| S24 | 4,232,495 | 921,369 | 0.9 | N/A | N/A | N/A | 10 |
| S25 | 4,234,639 | 921,337 | 0.6 | 3.3 | 1.9 | 1.1 | 27 |
| S26 | 4,234,362 | 922,710 | 1.5 | 3.3 | 1.9 | 1.1 | 20 |
| S27 | 4,231,288 | 921,546 | 1.6 | 6.3 | 2.5 | 1.0 | 20 |
| S28 | 4,234,481 | 924,999 | 1.9 | 5.6 | 3.0 | 1.6 | 26 |
| S29 | 4,233,906 | 922,691 | 1.9 | 4.0 | 2.2 | 1.2 | 28 |
| S30 | 4,233,200 | 924,451 | 0.0 | N/A | N/A | N/A | 24 |
| S31 | 4,233,063 | 923,358 | 0.9 | 3.6 | 2.1 | 1.2 | 27 |
| S32 | 4,232,411 | 922,703 | 1.8 | N/A | N/A | N/A | 27 |
| S33 | 4,231,969 | 924,106 | 1.7 | N/A | N/A | N/A | 31 |
| S34 | 4,230,155 | 925,059 | 0.8 | 5.4 | 3.2 | 1.9 | 23 |
| S35 | 4,228,883 | 925,090 | 1.0 | N/A | N/A | N/A | 7 |
| S36 | 4,228,696 | 923,460 | 1.6 | N/A | N/A | N/A | 9 |
| S37 | 4,228,105 | 924,045 | 2.1 | N/A | N/A | N/A | 11 |
| S38 | 4,227,490 | 923,386 | 1.9 | 4.0 | 3.3 | 2.7 | 28 |
| S39 | 4,227,375 | 924,408 | 1.6 | N/A | N/A | N/A | 29 |
| S40 | 4,228,228 | 922,430 | 1.0 | 4.0 | 2.5 | 1.6 | 30 |
| S41 | 4,233,658 | 923,826 | 0.3 | 9.8 | 2.4 | 0.6 | 26 |
| S42 | 4,219,971 | 396,474 | 1.3 | 6.1 | 3.5 | 2.0 | 29 |

Note: N/A is unavailable data.

### 4.1. Fundamental Frequency

It has commonly been assumed that the distribution of $f_0$ is closely related to geological conditions and depends mainly on shear wave velocity ($V_s$) and sediment thickness [38,39], which is related to the average vs. and inversely related to the sediment thickness. Figure 6

shows the distribution of the fundamental frequency of all stations, with $f_0$ being distributed between 0.8 and 9.8 Hz and gradually increasing from southwest to northeast. The sediment thickness in the area gradually becomes thinner from southwest to northeast, and the distribution of fundamental frequencies is inverse to the thickness layer. This indicates that the fundamental frequency values identified by the new method are consistent with the geological conditions, which indicates the potential applicability and reliability of the new method.

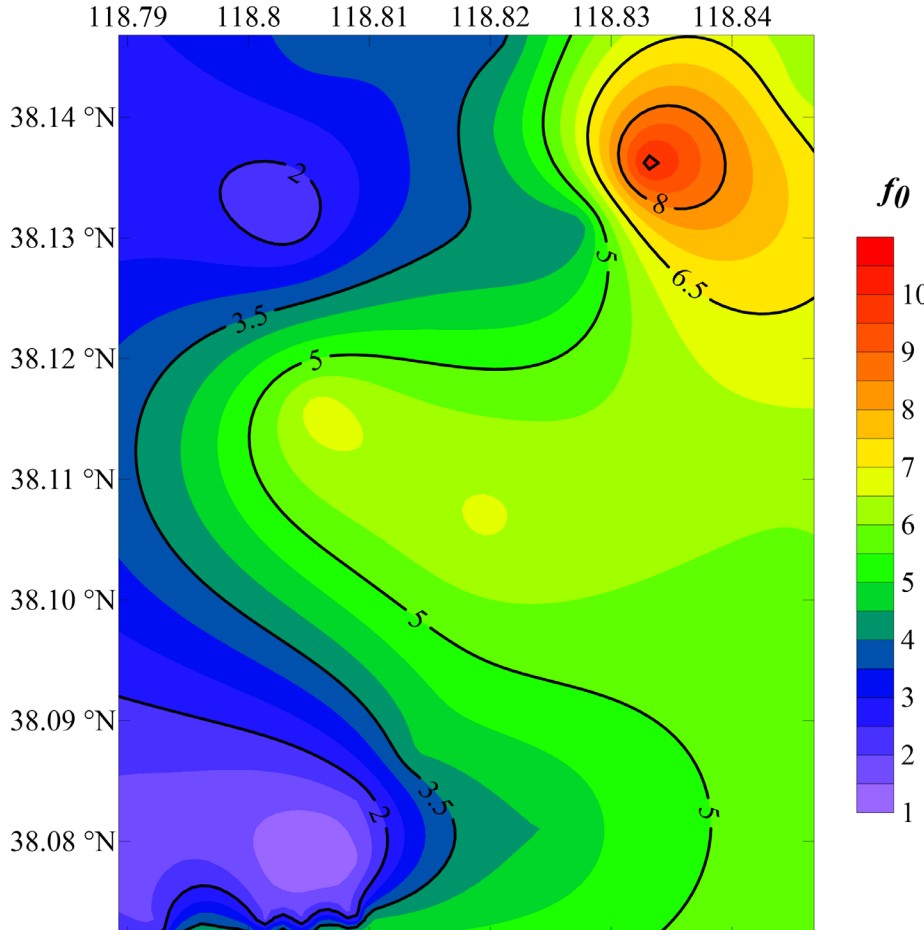

**Figure 6.** Distribution map of the fundamental frequency.

According to Figure 7, the stations in the BFE, ABE, and WISD areas have a stable distribution of the fundamental frequency values. It can be seen that the $f_0$ values of stations in BFE are lower than the others and basically stay between 1.3 and 1.8 Hz, while S3 has a higher $f_0$ of 3.7 Hz. This inconsistency may be related to the fact that this station is located on the back side of the seaward extending road, where the soils are less affected by wave-driven erosion and are more naturally consolidated than in other places. The $f_0$ values of stations in ABE are basically centered around 3.8 Hz, while the $f_0$ value at S23 is anomalous at 1.4 Hz. It is speculated that this $f_0$ value is low because this station had been inundated with water before the test, resulting in loose soils and shallow groundwater levels; thus, the wave velocity and amplification are weakened when the S-wave reaches the surface. The distribution of $f_0$ values in WISD ranges from 3.6 to 9.8 Hz, which ranges from middle to high levels.

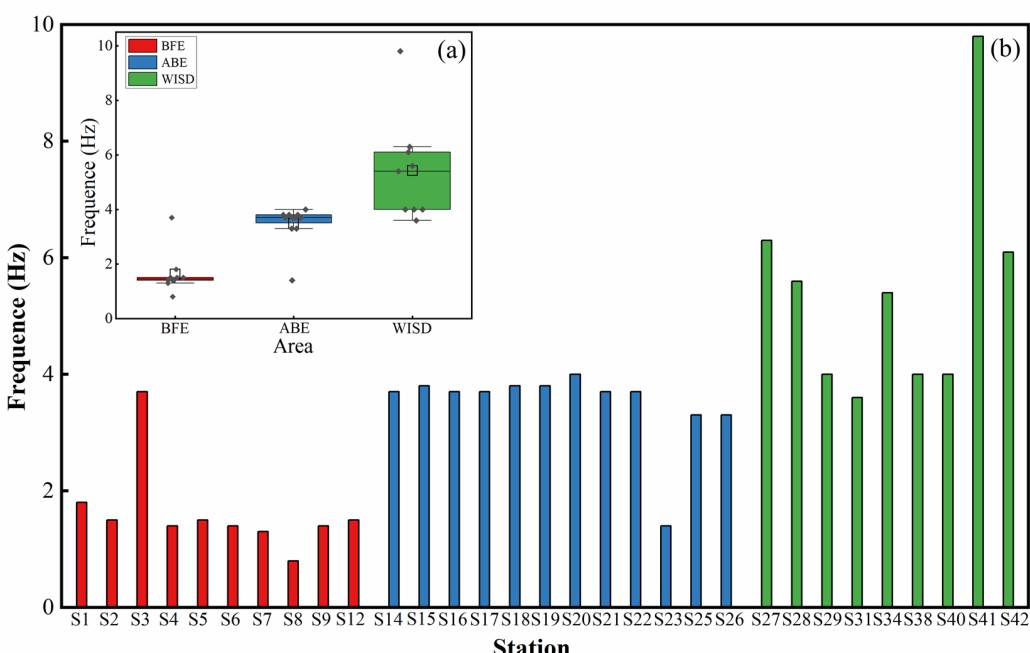

**Figure 7.** Fundamental frequency statistics for each station. (**a**) The distribution of fundamental frequency for stations in different study areas. (**b**) The fundamental frequency for all stations.

According to geological data, the study area is a flood plain established by the successive oscillations, migrations, and alluvial deposits of the Yellow River. Due to the effects of coastal currents and waves, thick deposits of loose sediments are widely distributed within the study area, with a gradual decrease in thickness from southwest to northeast. In addition, the stability of the embankment is undermined by the scouring and erosive actions of the waves and tides. These areas are more susceptible to ground motion amplification due to energy aggregation and the multiple reflections of seismic waves when an earthquake occurs, which will potentially cause substantial damage and possibly even liquefaction. The terrain behind the embankment slopes gently contains shallow groundwater, and the soil is basically saturated due to the poor drainage of groundwater and surface water. The stability of the soil behind the embankment is also low due to the frequent events of wave overtopping and scouring. The $f_0$ values in BFE and ABE are lower compared to those in WISD, or more precisely, to the soil on both sides of the embankment. Therefore, it is necessary to take precautionary measures regarding foundation treatments to prevent the embankment from being damaged by dynamic loads caused by strong storms or earthquakes.

### 4.2. Amplification

The peak amplitude, *A*, gives a rough indication of the difference in density between the surface sediments and the bedrock. The amplification of the seismic wave could occur due to significant differences in density between layers. In other words, when a seismic wave propagates from a high-density medium to a low-density medium, the amplitude of the seismic wave will increase. As shown in Figure 8, the amplification values are in the range of 1.8–3.5, which is in the medium to high amplification range. This indicates that the amplitude of the seismic wave will increase when it propagates through the overlying loose sediments in the area. From the HVSR curves of all stations, the peak amplitudes of the HVSR are mainly concentrated around three frequency bands, which are consistent with the distribution of fundamental frequencies in the BFE, ABE, and WISD regions. Additionally, "False peaks" are found in some curves in the lower frequency bands (0.2–0.4 Hz), which were mainly due to the disturbance caused by the turbulence of the wind near the geophone during observation. It has been reported that buildings can be damaged when amplification

exceeds a factor of 3 [28], and a total of 10 measurement stations recorded this amplification. Therefore, based on the distribution of the amplification, not only are both sides of the embankment potentially hazardous areas but there are also areas of vulnerability within the wetland.

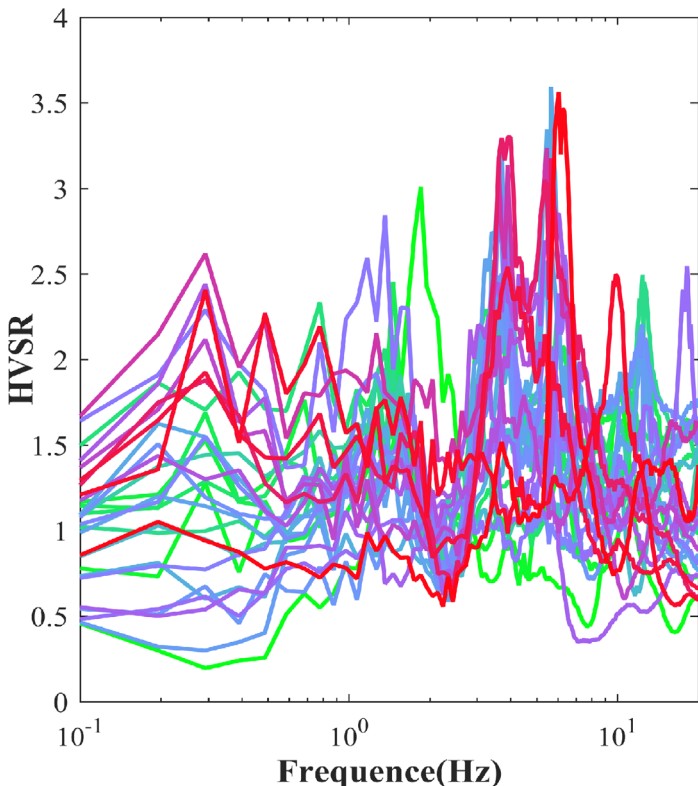

**Figure 8.** HVSR curves for all stations. Different colors represent different HVSR curves for different stations.

In addition, the multiple peaks in the HVSR curves also reflect the presence of multiple density interfaces in the stratum, indicating different impedance ratios at different depths [40,41]. These facts are consistent with the binary structure of the sediments in the Upper Delta Plain and the intercalated unevenly distributed soft soil layers. The different shapes of the HVSR curve reflect the lateral heterogeneity of the formation [42].

*4.3. Vulnerability Index*

The vulnerability index value can be considered an indicator of the degree of deformation of the soil [43], and it will help to detect weak points of the ground by comparing the $K_g$ values at different stations. Figure 9 shows that the distribution of $K_g$ ranges from 0.6 to 15.3, gradually decreasing from southwest to northeast, which is closely related to the local geological conditions. The $K_g$ values on both sides of the embankment are in the range of 1.1–15.3, while the $K_g$ values of the stations located on the wetlands are in the range of 0.6–2.7. Comparing Figures 6 and 9, it can be seen that the distribution of $K_g$ strongly coincides with $f_0$, for which a high vulnerability index is usually found in the areas with low fundamental frequencies. The stations in BFE have a higher vulnerability index due to the damage caused by scouring and hollowing or the impacts of siltation on the stability of the soil.

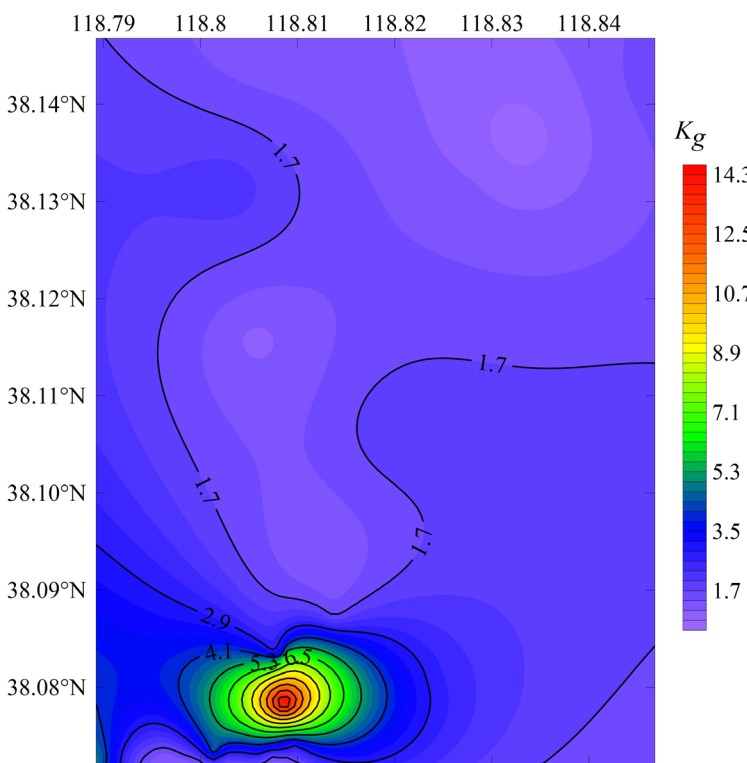

**Figure 9.** Distribution map of the vulnerability index.

The $K_g$ value is a qualitative description [44]. The higher the $K_g$ value is, the higher the liquefaction potential [45]. In the southwestern part of the study area, the stations on the beach area in front of the corner of the two embankments have the largest $K_g$ values, which are the weakest areas. It is more likely that the amplitude of the seismic motion will be amplified here due to the aggregation and multiple reflections of the energy of seismic waves; therefore, unstable soil is more likely to fail, and the liquefaction hazard is the highest. Therefore, it is necessary to adopt methods for treating foundations, such as replacement method and dynamic consolidation, to prevent the loose soil in front of the embankment from undergoing liquefaction or failures driven by uneven settlement during strong storms or earthquakes, which would result in damage to the embankment.

## 5. Conclusions

This work presents a case study evaluating the liquefaction potential of silt sediments based on the HVSR method. The ambient noise was recorded at multiple stations in the Yellow River Delta, and a new method was used to identify the fundamental frequency and amplification of the silt sediment. The results showed that $f_0$ was in the range of 0.8–9.8 Hz and $A$ was in the range of 1.8–3.5. According to the fundamental frequency and amplification, the vulnerability index was calculated to evaluate the liquefaction potential of the silt foundation. The following conclusions were drawn from this analysis:

1. This study presents a new method to identify the fundamental frequency from ambient noise recording in the Yellow River Delta. The results show that the soils in the areas with different geological conditions have a stable range of the fundamental frequency respectively, and the results are consistent with the inverse relationship between the fundamental frequency and the thickness of the sediment layer. Thus, the new method is potentially applicable and reliable for identifying the fundamental frequency;

2. The vulnerability indexes of the Yellow River Delta range from 0.6 to 15.3 and are dependent on the variability of the local geological conditions. In the southwestern part of the study area, the maximum value of $K_g$ occurs on the beach in front of the corner of the two embankments, where the liquefaction hazard is greatest; therefore,



the construction and maintenance of buildings should be carefully considered here. However, we suggest that combining the HVSR method with geotechnical investigation methods, such as field tests or indoor geotechnical tests, can further investigate the extent of liquefiable layers, which is essential for a more reliable assessment of the liquefaction potential of silty soils;

3. The results show that the HVSR method, when based on single station noise recordings, is suitable for rapidly assessing liquefaction potential. Compared with other field or laboratory tests, the HVSR method is a convenient, practical, faster, and non-destructive tool for assessing the liquefaction potential of silty sedimentary, such as the Mississippi River Delta. This will help to carry out liquefaction hazard assessment of silty soil in a new perspective, which can provide services for urban disaster prevention and mitigation and can also provide a reference basis for establishing regional seismic codes with good practical application value.

**Author Contributions:** Conceptualization and methodology, Q.M., Y.L., and W.W.; formal analysis, S.W. and W.W.; data curation, Y.L. and Y.C.; writing—original draft preparation, Y.L. and Q.M.; writing—review and editing, Y.L. and Q.M.; visualization, Y.C. and S.W.; project administration, Q.M.; funding acquisition, Q.M. All authors have read and agreed to the published version of the manuscript.

**Funding:** This research was funded by The National Natural Science Foundation of China (grant number 42272327) and the Social and livelihood project of Shandong Province (grant number 2021, 202131001).

**Institutional Review Board Statement:** Not applicable.

**Informed Consent Statement:** Not applicable.

**Data Availability Statement:** The data presented in this study are available on request from the corresponding author.

**Acknowledgments:** The authors would like to thank Zhiyuan Chen who participated in recording the ambient noise in the field and Yupeng Ren who participated in revising the manuscript.

**Conflicts of Interest:** The authors declare no conflict of interest.

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
