# Peer review of "A Case Study Assessing the Liquefaction Hazards of Silt Sediments Based on the Horizontal-to-Vertical Spectral Ratio Method"

_jmse, doi:10.3390/jmse11010104_

Round 1
Reviewer 1 Report
1) I think the calculation of the rate of subduction would help the reader understand the magnitude of the potential vulnerability
2) Should mention examples from around the world
3) Figure 4 and 7, bigger more comprehensible if possible
Author Response
Dear Reviewer:
Thank you for your precious comments concerning our manuscript entitled “A case study assessing the liquefaction hazards of silt sediments based on the horizontal to vertical spectral ratio method” (ID: jmse-2057491). Those comments are all valuable and very helpful for revising and improving our paper, as well as the important guiding significance to our research. We have studied the comments carefully and have made corrections which we hope meet with approval. Revised portions are marked in red on the paper. The main corrections in the paper and the responses to the reviewer’s comments are as flowing:
Point 1: I think the calculation of the rate of subduction would help the reader understand the magnitude of the potential vulnerability
Response 1: Thank you for your precious comment. The rate of subduction might be a good and interesting way to understand the magnitude of the potential vulnerability. However, I did not find any papers explaining the relevance of the rate of subduction to liquefaction hazard assessment. Furthermore, the difference is that the vulnerability in this paper is considered as an index to indicate the easiness of deformation of measured points. In this study, noise recordings were conducted at the surface to assess the liquefaction hazard of the surface silt sediments with the vulnerability index.
Point 2: Should mention examples from around the world
Response 2: Thank you for your professional suggestion. We have modified them in the revised manuscript (P1, 41-44).
Point 3: Figure 4 and 7, bigger more comprehensible if possible
Response 3: We gratefully appreciate your comment. We have corrected them in the revised manuscript (P11, 253; P14, 333).
We appreciate your warm work earnestly and hope that the correction will meet with approval. Once again, thank you very much for your comments and suggestions.
Best regards.
Yang Li
E-mail: liyang4009@stu.ouc.edu.cn
Corresponding author: Qingsheng Meng
E-mail: qingsheng@ouc.edu.cn
Corresponding author: Yang Li
E-mail: liyang4009@stu.ouc.edu.cn

Reviewer 2 Report
The manuscript is well-written and organized. Globally, the paper is sound and presents an interesting topic.
My alone comment, I suggest the authors add practical implications of the research with a focus on reproducibility and future works.
Author Response
Dear Reviewer:
Thank you for your precious comments concerning our manuscript entitled “A case study assessing the liquefaction hazards of silt sediments based on the horizontal to vertical spectral ratio method” (ID: jmse-2057491). Those comments are all valuable and very helpful for revising and improving our paper, as well as the important guiding significance to our research. We have studied the comments carefully and have made corrections which we hope meet with approval. Revised portions are marked in red on the paper. The main corrections in the paper and the responses to the reviewer’s comments are as flowing:
Point 1: The manuscript is well-written and organized. Globally, the paper is sound and presents an interesting topic. My alone comment, I suggest the authors add practical implications of the research with a focus on reproducibility and future works.
Response 1: Thank you for your positive feedback and precious suggestions on the present work. We have supplemented them in the revised manuscript (P15, 364-367). Modifications are as follows “This will help to carry out liquefaction hazard assessment of silty soil in a new perspective, which can provide services for urban disaster prevention and mitigation, and can also provide a reference basis for establishing regional seismic codes with good practical application value.”
We appreciate your warm work earnestly and hope that the correction will meet with approval. Once again, thank you very much for your comments and suggestions.
Best regards.
Yang Li
E-mail: liyang4009@stu.ouc.edu.cn
Corresponding author: Qingsheng Meng
E-mail: qingsheng@ouc.edu.cn
Corresponding author: Yang Li
E-mail: liyang4009@stu.ouc.edu.cn

Reviewer 3 Report
The liquefaction hazard represents a serious worldwide problem and is the direct responsible for human and economic loss. The authors proposed an interesting case study to assess the liquefaction hazard in the Yellow river delta. They introduce a new methodology to evaluate the soil's fundamental frequency and calculate a vulnerability index.
The paper is clear, readable, well-organized, and written. The abstract and introduction sections are well-focused on the paper topic.
Unfortunately, the paper is missing crucial information. The authors should be provided a better description of the proposed methodology; shows more results from the different stations and studying area, and compare the results with others studies or the references methodologies.
Personally, the paper is valid and interesting and with more data and a section dedicated to comparing the obtained result with others studies the final results will be more robust and valid from the scientific point of view.
More in detail:
I suggest improving the methodology section with a general figure about the Fourier amplitude spectrum ed information about the HVSR and Kg range of values and interpretation.
In section 3.2 the information about the acquisition time duration is missing (is 160s as reported in figure 3?). It is the same for all the stations?
Section 3.3 needs to be improved. Please provide a flow chart or figure to show the approach and results. How many stationary windows of 10s did you analyze for each station? Could you organize a table to report all the crucial info?
The discussion section needs to be provided with more info: show the results of the other sites and compare them with other studies or methodologies.
Lines 20-23 Please check the sentence for typo
Line 64 With “cultural origins” do you means Anthropic activities?
Line 98 all the components of the Fourier amplitude spectrum is measured at the surface. Please modify the sentence
Equation 2 please specify the vulnerability index (Kg) with more details (means of g)
Figure 1c I suggest using a different color for the three zones to make it more readable
Figure 3a the scale is missing a color legend (only the white color is explained in the main text)
Figure 3c Title should be H/V, not V
Line 172-173 The proposed methodology should be explained and detailed much better (it is representing the paper's core and novelty point). Moreover, should be compared with the referenced one.
Line 183-183 Check the citation order of figure 3c and figure 3b probably they are inverted
Line 206-208 Could be useful to show a map of Vs and sediment thickness distribution for the studying area
Figure 4 If possible, improve the figure with the target site location.
Line 202 Could you provide an example of a data station not met the SESAME criteria?
In the present form, the paper is eligible for publication on JMSE only after major revision.
Author Response
Dear Reviewer:
Thank you for your precious comments concerning our manuscript entitled “A case study assessing the liquefaction hazards of silt sediments based on the horizontal to vertical spectral ratio method” (ID: jmse-2057491). Those comments are all valuable and very helpful for revising and improving our paper, as well as the important guiding significance to our research. We have studied the comments carefully and have made a correction which we hope meets with approval. Revised portions are marked in red on the paper. The main corrections in the paper and the responses to the reviewer’s comments are as flowing:
Point 1: I suggest improving the methodology section with a general figure about the Fourier amplitude spectrum ed information about the HVSR and Kg range of values and interpretation.
Response 1: We gratefully appreciate your comment. We have improved the figure about the Fourier amplitude spectrum (P2, 93-95; P3, 100-103), the information about the HVSR (P3, 104-109), and the Kg (P3, 111-115) in the revised manuscript.
Point 2: In section 3.2 the information about the acquisition time duration is missing (is 160s as reported in figure 3?). It is the same for all the stations?
Response 2: Thank you for your valuable suggestions. We are very sorry for our negligence of the acquisition time duration. We have added the information in the revised manuscript (P5,151-152). Furthermore, it is same for all the stations with the acquisition time duration.
Point 3: Section 3.3 needs to be improved. Please provide a flow chart or figure to show the approach and results. How many stationary windows of 10s did you analyze for each station? Could you organize a table to report all the crucial info?
Response 3: We sincerely appreciate the valuable comments. We have improved a flow chart of the approach in the revised manuscript (P6, 169-171).
Moreover, we have added the results of ambient noise recording and the number of stationary windows for each station in Table 1 in the revised manuscript (P9, 240).
Point 4: The discussion section needs to be provided with more info: show the results of the other sites and compare them with other studies or methodologies.
Response 4: We gratefully appreciate for your suggestions. We have provided the results of ambient noise recording in Table 1 (P9, 240). As we mentioned in the paper, it is equally expected that the combination of HVSR methods with geotechnical investigation methods, such as field tests or indoor geotechnical tests, which is essential for a more reliable assessment of the liquefaction potential of silty soils. However, it is very unfortunate that there are no other relevant studies in this study area.
Point 5: Lines 20-23 Please check the sentence for typo
Response 5: We sincerely thank you for your careful reading. As suggested by the reviewer, we have corrected them in the revised manuscript (P1, 20-23).
Point 6: Line 64 With “cultural origins” do you means Anthropic activities?
Response 6: Thank you for your review to our manuscript. I'm truly sorry that I didn't express it properly here. Yes, “cultural origins” means anthropic activities, and the noise has basically two different origins: natural or cultural.
Point 7: Line 98 all the components of the Fourier amplitude spectrum is measured at the surface. Please modify the sentence
Response 7: Thanks for your careful checks. We are sorry for our carelessness. Based on your comments, we have modified them in the revised manuscript (P3, 98).
Point 8: Equation 2 please specify the vulnerability index (Kg) with more details (means of g)
Response 8: We gratefully appreciate for your carefully pointing out the inadequate description of the details. We have added them in the revised manuscript (P3, 111).
Point 9: Figure 1c I suggest using a different color for the three zones to make it more readable
Response 9: We sincerely appreciate the valuable comments. As suggested, we have corrected it in the revised manuscript (P4, 130).
Point 10: Figure 3a the scale is missing a color legend (only the white color is explained in the main text)
Response 10: We sincerely thank you for the comment. Active windows are displayed in color, while discarded windows are white. Here the color has no special meaning. It is to distinguish the different active windows, which can visualize the windowing more intuitively. Therefore, we have not added a color legend to the figure.
Point 11: Figure 3c Title should be H/V, not V
Response 11: We were really sorry for our careless mistakes. Thank you for your reminder. We have corrected it in the revised manuscript (P8, 212).
Point 12: Line 172-173 The proposed methodology should be explained and detailed much better (it is representing the paper's core and novelty point). Moreover, should be compared with the referenced one.
Response 12: Thank you for your precious comment. We have added the explanation and details of the proposed methodology in the revised manuscript (P7, 188-195).
Furthermore, we sincerely appreciate your valuable suggestion. As suggested by the reviewer, we have complemented the explanation and a figure to illustrate the results compared with the referenced one (P8, 217-233).
Point 13: Line 183-183 Check the citation order of figure 3c and figure 3b probably they are inverted
Response 13: Thank you for your precious reminder. We were really sorry for our careless mistakes. We have corrected it in the revised manuscript (P7, 199-201).
Point 14: Line 206-208 Could be useful to show a map of Vs and sediment thickness distribution for the studying area
Response 14: We gratefully appreciate for your comment. The distribution of mean Vs and sediment thickness in the study area could be useful for the explanation of the fundamental frequency, but it is unfortunate that the shear wave velocity test was not performed during the field tests. Nevertheless, we have obtained the distribution of sediment thickness in this study area by reviewing information and relevant data from boreholes, which is briefly described in 3.1 Geologic setting.
Point 15: Figure 4 If possible, improve the figure with the target site location.
Response 15: Thank you for your precious suggestions. Adding the target site location to the figure will give the reader a clearer understanding of the fundamental frequency of different sites, but it will also have an impact on reviewing the distribution of the fundamental frequency of the whole area. Therefore, we did not add the target site location to the figure in the manuscript.
Point 16: Line 202 Could you provide an example of a data station not met the SESAME criteria?
Response 16: We sincerely appreciate your valuable comments. Here we provide data of S11, which is not meet the SESAME criteria. The number of active windows for the data is 8 after windowing, which cannot meet the minimum number of windows (10). It does not guarantee the reliability of the observations. This may be caused by poor coupling of the geophone to the soil during the acquisition.
We appreciate for your warm work earnestly, and hope that the correction will meet with approval. Once again, thank you very much for your comments and suggestions.
Best regards.
Yang Li
E-mail: liyang4009@stu.ouc.edu.cn
Corresponding author: Qingsheng Meng
E-mail: qingsheng@ouc.edu.cn
Corresponding author: Yang Li
E-mail: liyang4009@stu.ouc.edu.cn

Round 2
Reviewer 3 Report
The authors revised the manuscript follows indications and suggestions proposed. In my opinion, the paper quality has been improved and worth to be published in this form.